# Stress-Inducing Factors vs. the Risk of Occupational Burnout in the Work of Nurses and Paramedics

**DOI:** 10.3390/ijerph19095539

**Published:** 2022-05-03

**Authors:** Aneta Grochowska, Agata Gawron, Iwona Bodys-Cupak

**Affiliations:** 1Department of Nursing, Faculty of Health Sciences, University of Applied Sciences in Tarnow, 33-100 Tarnów, Poland; 2ICU, Health Care Centre in Pinczow, 28-400 Pińczów, Poland; aga1966@interia.pl; 3Laboratory of Theory and Fundamentals of Nursing, Faculty of Health Sciences, Institute of Nursing and Midwifery, Jagiellonian University Medical College, 31-126 Krakow, Poland; i.bodys-cupak@uj.edu.pl

**Keywords:** nurses, paramedics, burnout, stress

## Abstract

Introduction: Contemporary healthcare faces new challenges and expectations from society. The profession of a nurse, as well as a paramedic, is essential for the efficient functioning of healthcare. It has its importance not only in promoting and preserving health but also in prevention. With the increasing importance of providing medical care at the highest level, it is expected of these two professional groups to have more knowledge and skills than a few years earlier. The daily contact with patients and their families, the low level of control of the environment, the hierarchical system of professional dependence, and the dissatisfaction with remuneration are becoming extremely burdensome aspects of the nursing and paramedic professions. Long-term exposure to stressors associated with these medical professions may, in the long term, lead to the emergence of occupational burnout syndrome. The aim of this study is an attempt to answer the question of whether and how stress factors affect the occurrence of occupational burnout in the work of nurses and paramedics working in various medical entities. Material and Methods: The study covered a group of 434 respondents, including 220 nurses and 214 paramedics, working professionally in hospital departments and care and treatment facilities as well as in hospital emergency departments and ambulance services. The study was carried out using a diagnostic survey based on the questionnaire technique using the authors’ questionnaire and the standardized MBI Ch. Maslach. Two statistical values were used to statistically analyze the research results and verify the adopted hypotheses: the chi-square test and the Student’s *t*-test. Results and Conclusions: The current study showed that the phenomenon of occupational burnout among the studied group affects only nurses, while this problem does not apply to the studied paramedics. The main stressor among the nurses and paramedics is, above all, a very high level of responsibility. Nurses are overburdened by excessive demands and shift work, while paramedics are mostly burdened by an excess of duties. Both nurses and paramedics claim that their work is often stressful, which leads to physical and mental exhaustion.

## 1. Introduction

Professional work is a crucial part of the functioning of a modern person. However, it contributes to a wide range of frustration levels, which affects health in a negative way [1,2]. The recent civilization progress and resulting technological developments have been causing constant changes in the working conditions and in the organization of workplaces, triggering high levels of stress [3].

Stress is a stimulus and the reaction to it occurs on various levels: physical, behavioral, emotional, and cognitive [4]. The nurse is an important employee in the labor market of medical services, with both the patients and their families being the subjects of their professional activities [5]. Factors that may cause a psychological imbalance in a nurse, and lead to occupational burnout, may be associated with the working conditions and organization, the patient and his family, and the relationship with the therapeutic team, as well as discomfort resulting from the profession [6,7,8]. Stress can be experienced by any human being. Burnout, however, affects those people who show strong motivations and increased expectations related to their professional work [6].

The problem of occupational burnout has been considered in the literature on psychology since the 1970s. The very word “burnout” is metaphorical, but it accurately reflects the essence of exhaustion that is experienced by an individual as a result of highly stressful work conditions [9]. The speed of development of occupational burnout is the consequence of the tasks that are still imposed on the nurse, and of her subjective feeling of emotional exhaustion [6]. Working as a nurse or a paramedic is inherently associated with experiencing stress [10]. For a nurse, work can be a source of professional satisfaction as well as an excessive burden. This job involves a high risk of occupational burnout syndrome due to the occurrence of chronic stress [6,11,12].

The problem of stress occurring at work and the burnout associated with it is a serious social issue. It was noticed several decades ago, but its importance is still underestimated. The stress response varies from person to person. Coping with stress may be more or less effective depending on the method we use and a specific situation [13].

Occupational burnout occurs especially in professions in which close and committed interaction with another person is the basis of professional activity. It determines success and development in a specific profession. The effects of this intimate relationship, such as being familiar with suffering, chronic stress, and negative emotions, appear when the nurse cannot cope with the lack of success, professional burdens, and failures. Despite efforts, it seems impossible to find a way out of burdens and one suffers from emotional exhaustion and extreme tiredness [11]. In the work environment of nurses, there is a wide range of factors that are a source of stress and affect the psychological burden and occupational burnout [14,15]. Therefore, there is no doubt that it is stress that causes burnout [16].

Healthcare workers are indeed one of the occupational groups which are the most exposed to this unfavorable phenomenon. Their work requires a lot of professional commitment, which causes a great deal of physical and mental exhaustion. This group is exposed to various factors that trigger chronic stress syndrome. Both nurses and paramedics are in a close relationship with other people and their suffering, experiencing chronic stress and negative emotions. These effects occur when nurses and paramedics cannot cope with the lack of success, workload, and failures. In their work, they are constantly involved in situations that force them to experience professional stress, the main reason of which is constant contact with patients, who continually expect care and help from them.

This paper is dedicated to the topic of stress-inducing factors and the risk of occupational burnout of nurses and paramedics.

The aim of the research was an attempt to answer the question of whether and how stress-inducing factors in the work of nurses and paramedics affect the occurrence of occupational burnout among them during the pandemic situation. Due to its nature, a pandemic situation is a strong source of stress for people. Medical personnel experience high levels of stress, anxiety, and sleep disorders [17,18].

## 2. Material and Method

The paper implements the method of a diagnostic survey, using a questionnaire as a research technique. The tool used was the authors’ questionnaire and the questionnaire on occupational burnout by Ch. Maslach.

The authors’ questionnaire consisted of 30 questions that were created in order to determine whether the stressful nature of the work of nurses and paramedics influences their private lives, how they cope with stress at work, and whether they feel financial fulfilment and job satisfaction. The purpose of these questions was to consider the impact of stress-inducing factors on burnout affecting this particular professional group.

The Maslach Burnout Questionnaire was developed by Maslach and Jackson in 1981. The questionnaire consists of two parts. The first part contains 20 questions that have been divided into three groups, and each of these groups deals with one of three elements of occupational burnout. Questions from 1 to 9 concern emotional exhaustion, questions from 10 to 14 verify symptoms related to depersonalization, and questions from 15 to 20 concern professional satisfaction. The questionnaire also includes two additional off-scale questions which concern achievements, goals, and treatment of emotional problems at work. The second part of the questionnaire contains questions related to financial satisfaction. The respondents’ results are calculated separately for each of the subscales, according to the key. In order to calculate the risk of occupational burnout, positive answers from groups I and II and negative answers from group III need to be added. The results obtained in the last scale should be interpreted by taking into account the dimensions signifying occupational burnout: a low level of professional satisfaction means a high level of burnout, and a low level of satisfaction means the opposite of the burnout syndrome.

The questions are structured so that it is possible to establish descriptive data regarding the nurses and paramedics who participated in the tests. There were questions concerning gender, age, marital status, education, professional experience, work system, and place of residence.

### 2.1. Organization and the Course of Conducted Research

The study was conducted between November 2020 and January 2021. The number of people who participated in the research is 434. They all work professionally in hospital departments and care and treatment facilities, as well as in hospital emergency departments and ambulance services. The questionnaire was provided in electronic form on the Google Forms platform. The time to complete the questionnaires was irregular. One of the most important issues during the survey was its anonymity. All questionnaires contained complete responses, so it was not necessary to reject them.

### 2.2. Statistical Analysis Methods

The differences between the variables were verified using the Pearson chi square test and the Student’s *t*-test. Both research tools are used to compare two groups: dependent and independent variables. The significance level was *p* < 0.05. The calculations were performed with IBM SPSS Statistics 20 program.

### 2.3. Characteristics of the Study Group

A total of 434 people involved in health services participated in the survey. The questionnaire was completed by 220 (51%) nurses and 214 (49%) paramedics. There were 232 women and 202 men in total among the respondents. The majority of women are nurses, and men mostly work as paramedics. This suggests that gender is a factor when it comes to analyzing the professions. The respondents come from cities and villages (Table 1).

The respondents are from 20 to 60 years old, with some years of experience in the profession. Most of the nurses work in two jobs (Table 2).

### 2.4. Ethical Considerations

Nurses and paramedics were informed of the confidentiality and anonymity of the study, that their participation was voluntary, and that they may cease to cooperate at any time during the study.

The study was conducted in accordance with the principles of the Helsinki Declaration.

## 3. Results

Job satisfaction largely depends on the salary received. The respondents (88.2% of nurses and 85% of paramedics) almost unanimously stated that they are not paid adequately for the work that they perform (Figure 1). Only 52 people out of all respondents are satisfied with their salary. This result is low. The sense of obtaining an adequate salary builds a positive perception of professional work. Only 14.8% of the respondents stated that they could not say whether they were adequately remunerated in relation to their duties.

Psychological help is important for people whose professions are stressful and responsible as it helps to relieve stress. However, in both groups of respondents, in general, only 7.4% of the respondents used professional psychological assistance. This is a small percentage. The others unanimously stated that they do not use the services of a psychologist (Figure 2).

More than half of the nurses could not state explicitly whether the problem of occupational burnout concerned their group (57.2%). Among the paramedics, 31.8% of respondents could not assess whether they suffer from burnout syndrome. On the other hand, it was the paramedics (38.3%) who admitted that they noticed symptoms of burnout in themselves. Compared to the results obtained in the group of nurses, this was a significant percentage of affirmative responses (Figure 3).

### 3.1. Respondents’ Level of Occupational Burnout

The results of the study obtained by using Ch. Maslach’s burnout questionnaire are presented in Table 3 and Table 4. Firstly, a summary of the results concerning nurses and paramedics is presented with reference to the basic dimensions: professional exhaustion, depersonalization, and lack of a sense of professional achievement.

The overall score indicates that nurses show a risk of occupational burnout.

After summing the affirmative responses from the EW (occupational exhaustion) and DP (depersonalization) groups and the negative responses from the ZO group II (lack of sense of occupational achievement), an overall score was obtained. The higher the score, the greater the phenomenon of occupational burnout. The results obtained in the group of the surveyed paramedics indicate that they do not manifest symptoms of occupational burnout.

In addition to the burnout scale, two questions were added regarding the financial satisfaction with one’s job. The data obtained show that neither nurses nor paramedics are satisfied with their salaries (Table 5). They believe that their work should be better compensated. An overwhelming number of nurses and paramedics believe that their salaries are not adequate for the duties they perform. The salary does not reflect the work overload and commitment to the duties performed.

### 3.2. The Level of Occupational Burnout and Selected Variables

Based on the data in Table 6, it is clear that paramedics in their own assessment are more affected by professional burnout than the nurses surveyed.

However, there is no correlation between the occupation—nurses, paramedics—and feelings of depersonalization and emotional burnout. The results of the analysis showed that the *p*-value was only slightly higher than the assumed significance level of 0.05 and was 0.6. Hence, there was no significant relationship between the variables (Table 7).

The analysis using Student’s *t*-test proves that there is a correlation between the level of occupational burnout and the performed profession. A slightly higher score was obtained by nurses, so it can be concluded that the level of occupational burnout was higher in this group (Table 8).

The analysis of the data allowed for concluding explicitly that there is a relationship between the length of their professional experience and the frequency of perceived sense of physical exhaustion of nurses. Based on the results, it can be concluded that there is no relationship between work experience and physical exhaustion of paramedics (Table 9).

On the basis of data analysis by Pearson’s chi square test, it can be concluded that the *p*-value is higher than 0.05, and reached 0.57. Therefore, it can be determined that the amount of remuneration received for professional work is not more important for nurses than for paramedics (Table 10).

Based on the data analysis, it can be concluded that the *p*-value is higher than 0.05, and reached 0.53. Therefore, it can be determined that there is no relationship between the salary adequacy of nurses and paramedics (Table 11).

Moreover, the statistical analysis proved that there is no correlation between the use of psychological help and the indication of death as a stress factor in the professional work of both groups of respondents. The *p*-value is significantly higher than the assumed significance level, and reached 0.93 (Table 12).

## 4. Discussion

Today’s healthcare is facing new challenges and expectations from society. The nursing profession, as well as the paramedic profession, is essential to the efficient functioning of healthcare. It has its importance not only in promoting and maintaining health, but also in prevention. With the increased importance of providing the highest level of medical care, more and more knowledge and skills are expected of these two occupational groups than a few years earlier. Yet, it is the daily contact with patients and their families, the low scrutiny of the environment, the hierarchical system of professional dependency, and the dissatisfaction with salary that have become extremely burdensome aspects of the nursing and paramedic profession. Prolonged exposure to the stressors associated with these medical professions over time may lead to the onset of burnout syndrome. During the pandemic period, when these studies were conducted, the occurrence of occupational burnout was very real and particularly severe, as has been shown by other studies [19,20,21,22].

The first studies dealing with the issue of burnout among the medical profession began to be published in the United States in 1974. The scale of this phenomenon has given rise to a lot of research and the appearance of thousands of publications. These are theoretical works, where an attempt was made to define the concept of occupational burnout, as well as empirical ones, thanks to which materials were provided to find answers to questions concerning the causes, symptoms, effects, and possible ways to prevent stressors and the phenomenon of burnout.

The results collected on the basis of Ch. Maslach’s questionnaire for the present study prove that it was the nurses who showed the occupational burnout syndrome, while it was not registered in paramedics. Our own observations indicate that the tool prepared by Ch. Maslach is the most frequently used test to measure the occurrence of professional burnout among healthcare workers. Nurses and paramedics are some of the groups which are most vulnerable to occupational burnout.

Wilczek-Rużyczka and Zaczyk reached similar conclusions to the ones drawn in the current study. Their research also shows that nurses working in Poland suffer from professional burnout. This is indicated by the medium level of intensity of emotional exhaustion and depersonalization and a medium or even high level in terms of reduced sense of personal accomplishment [23].

The research among nurses by Kurowska and Zuza-Witkowska based on Ch. Maslach’s questionnaire also suggests that this professional group is exposed to occupational burnout. The study showed a moderate tendency to occupational burnout, but the authors of this scientific study also pointed out that the intensity of this phenomenon depends on the nature of work. The most vulnerable groups were found to be nurses working in a clinic, distinguished by the highest average age. The fewest symptoms of burnout were observed in nurses working in the noninvasive wards, whose average age is similar and the length of service in the hospital is relatively short [24].

However, the group of researchers who prepared a review of the literature on occupational burnout in the group of nurses note that the problem does not affect only those working in this profession in Poland. Research conducted with the Copenhagen Burnout Inventory by Hasselhorn shows that the highest level of occupational burnout was observed among nurses working in France and Slovakia, while the lowest was in the Netherlands. The Polish nurses in the current study are in the group of average results together with the Italian and German representatives [25].

The nurses who participated in the current study indicated that the most important stress factors in their work included, above all, very high responsibility, excessive demands, and shift work. Studies conducted by Kędra and Nowocień also led to similar conclusions. The most frequently indicated stress factors in the work of nurses are excess of duties and responsibility for another person’s health. On the other hand, an overload of professional duties is a significant factor causing a loss of job satisfaction and leading to occupational burnout. Similar conclusions were drawn by a research team from the Jagiellonian University. According to their research, the nurses studied were exposed to stress in their professional lives on a daily basis. The pressure of being responsible for the health and life of another person and unsatisfactory salary were considered to be its source. The authors also emphasize that the demanding attitudes of patients’ families contribute to the lack of respect for their work [10,26].

Studies on the effects of long-term stress among nurses very often show sleep problems and insomnia. Ogińska and Pokorski indicate that almost half of the nurses suffer from sleep deficiency caused by shift work. Migraines, neurosis, and gastrointestinal problems were also frequently found to be equally important consequences of long-term exposure to stress factors among nurses [27].

Similar conclusions were drawn based on the current research for this study. Among the nurses surveyed, the most common symptoms of stress were chronic headaches and emerging irritability and anxiety.

The analysis of the current research based on the answers from the Ch. Maslach questionnaire submitted by the paramedics proved that the occupational burnout phenomenon does not occur in the studied group, although the result was close to the low level. However, different results can be observed in the work of Satan and Harazin. Among their respondents, the phenomenon of occupational burnout occurs in all its dimensions (emotional exhaustion, depersonalization, and the sense of own achievement). Additionally, the researchers noted that occupational burnout is influenced by gender, age, personal situation, and work experience. At the same time, no correlation was found between the profession, the level of education, and the burnout process.

Similarly, research conducted by Piekarczyk indicates that the occupational burnout syndrome concerns nurses to a moderate degree. However, the author stresses that this phenomenon has a gradual character. A large group of respondents does not have a fully-symptomatic occupational burnout syndrome: only one or two of the three dimensions are significantly increased [28,29].

The research undertaken by a team of researchers from the Krakow and Warsaw environments also stresses that the paramedic profession is one of the professions where exposure to stress is particularly high and there is a high risk of developing burnout syndrome. However, the research results of this team are consistent with those obtained from the current study. The analysis of data provided by paramedics working in Krakow and its surroundings shows that 35% of the respondents noticed the feeling of fatigue and tiredness, which they identified as a symptom of occupational burnout [30]. Similar results were found in the present study; more than 38% of the examined paramedics in their own assessment felt the symptoms of occupational burnout syndrome. Almost the same results were obtained by a team of Israeli researchers in a group of paramedics, where occupational burnout occurred in 35% of the respondents [31]. Thus, the results are comparable on a global scale.

Occupational burnout is influenced by stressors that occur continuously. Among the group of paramedics surveyed for the present study, the most frequently mentioned stressors include excessive duties, exposure to harmful factors, and too-high responsibility. Additionally, the respondents indicated that dissatisfaction and resentment of patients and their families are particularly burdensome in their work. The research team of Ślusarska, Nowicki, and Jędrzejewicz came to interesting, yet similar, conclusions. Their research concluded that in the work of paramedics, the most significant stress factor is the presence of child victims, as well as the threat to their own and their colleagues’ lives. Among other stressors, the authors also identified fear of losing one’s job, pressure from superiors, and the demanding nature of patients. Additionally, night work and shift work are perceived as stressors [32]. These responses are also reflected in the current study.

In Buljan’s study, paramedics are at risk of occupational exhaustion syndrome, which results in a reduced sense of security in the workplace. The risk factors that influence the occupational exhaustion syndrome are the uncertainty of the system’s efficiency, the feeling of psychological workload caused by work, poor social contacts resulting in a lack of help from others, and the lack of positive motivators at work, such as various rewards [33]. These studies are also supported by the current research.

The comparison of the group of nurses and paramedics in terms of the phenomenon of occupational burnout was also discussed by Nowakowska and Wolniewicz. Among the representatives of both professional groups, common factors contributing to the development of occupational burnout syndrome were listed. They included, among others, passivity and uncertainty in contacts with people, lack of support and recognition from the employer, and a high mental strain connected with responsibility. The authors of the publication also assembled the symptoms of burnout that could be observed in everyone, that is, a high baseline for emotions, a dimension of exhaustion, a measure of depersonalization, and a low score for a sense of achievement. Summarizing the results of the Ch. Maslach’s questionnaire proved that both Nowakowska’s and Wolniewicz’s research groups experienced occupational burnout syndrome [34]. These results are in contrast to those presented in the present study, where it was only the nurses who suffered from syndrome.

The studies on occupational burnout among nurses in the world confirm the results of the present study. In the vast majority of nurses, there is a systematic increase in the appearance of occupational burnout syndrome. Occupational burnout affects more than one-third of nursing staff in Iran. Therefore, effective interventions and strategies are needed to reduce and prevent occupational burnout among nurses [35]. Meanwhile, a study undertaken in China found that nurses there who had higher self-esteem traits reported less emotional exhaustion, which translated into higher professional effectiveness. The authors concluded that improvement of coping strategies may be helpful in preventing occupational burnout among nurses, thus increasing professional effectiveness [36].

The results of the study presented by Nirel et al. suggest that the pressure present in the work of paramedics may be caused by lack of administrative support, long working hours, lack of work–life balance, and insufficient remuneration. Respondents are unable to cope with responsibilities, the pressure of working in precarious conditions, and rapid changes from calm situations to emergencies. Job dissatisfaction is caused by burnout, work overload, and poor health. Physical and mental health that limits their ability to work is associated with feelings of burnout and a desire to change jobs [31].

The discussion indicates the legitimacy of the conducted research and is an implication for planning further analyses. It is necessary to broaden the research area, as well as focus on specific, narrowly specialized areas of medical care. In order to establish preventive recommendations to avert the development of the occupational burnout syndrome in specific groups of medical workers, it is necessary to precisely identify recipients of these recommendations. Therefore, research must be carried out in homogeneous groups, e.g., among paramedics, nurses, or emergency medicine doctors.

The comparison of occupational burnout and stressors occurring in the work of paramedics and nurses is extremely important to understand the specificity of their professional work. Each of these groups working in the health service is exposed to different stressors, which is reflected in the essence of the occurrence and level of burnout. Both researched groups experience excessive mental strain, which may affect competence and quality of work, as well as disturb their private lives. Considering the current research and a review of other scientific studies on stress and burnout among nurses and paramedics, special attention should be paid to the program of activities counteracting burnout and excessive stress. Above all, it is the improvement of working conditions and an increase in salaries that will motivate both groups to continue working. In addition, it would be advisable to introduce widely available training and workshops on how to cope with stress and high mental strain, as well as to offer a widely available range of psychological support available. This would benefit not only the healthcare professionals mentioned in the study, but also the patients, their families, and the entire healthcare system.

## 5. Conclusions

Both nurses and paramedics claim that their work is very stressful, which leads to physical and mental exhaustion.

According to the self-assessment, it is the paramedics who claim to feel the symptoms of occupational burnout.

There is a correlation between the appearance of the syndrome of occupational burnout and the work performed in the study groups.

The work experience of nurses influences physical exhaustion. Such a correlation was not observed in paramedics.

None of the studied groups is adequately remunerated in relation to their duties. The level of remuneration is no more important for nurses than for paramedics.

### Limitation of the Study

The main limitations of the study concern data collection methods. Data collection was carried out at a certain point in time, not longitudinally. Another limitation may be the selection of the group of respondents. The comparison of the occurrence of occupational burnout among nurses working in different departments and paramedics may cause inaccuracies. Certainly, it is necessary to conduct further research, taking into account the selection of the group in terms of the specificity of the performed professional tasks, e.g., emergency nurses and paramedics.

## Figures and Tables

**Figure 1 ijerph-19-05539-f001:**
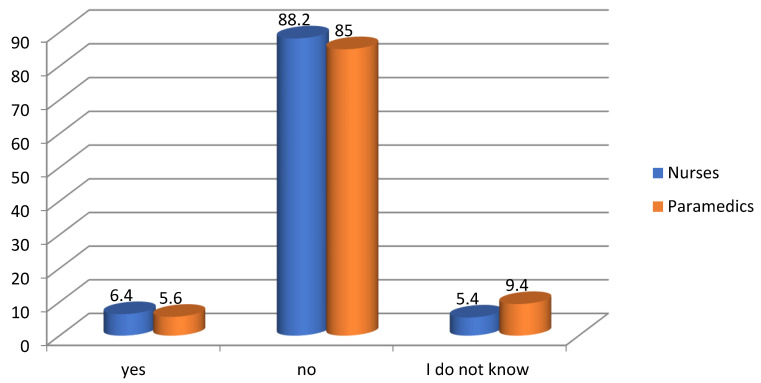
Respondents’ opinions on the adequacy of salary in relation to the work they perform.

**Figure 2 ijerph-19-05539-f002:**
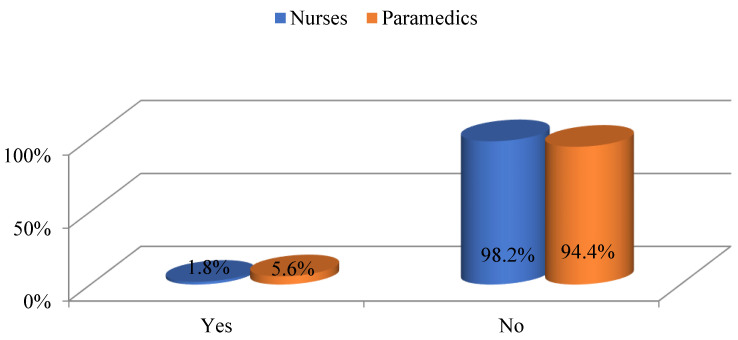
Percentage of nurses and paramedics using psychological help.

**Figure 3 ijerph-19-05539-f003:**
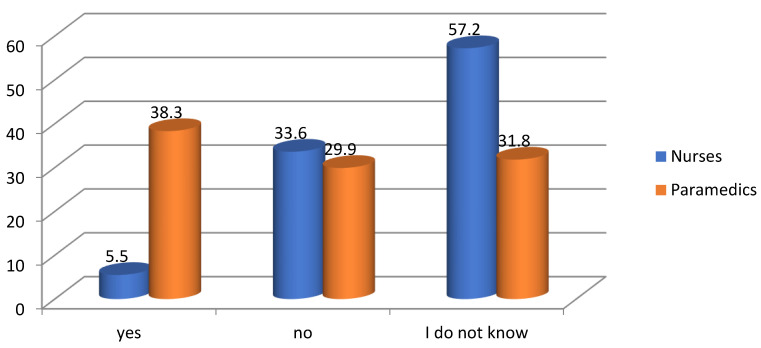
Results of self-assessment of the occurrence of occupational burnout among the respondents.

**Table 1 ijerph-19-05539-t001:** Demographic characteristics of the respondents. (N—nurses, P—paramedics).

Variable		Nurses		Paramedics	
		N	%	N	%
Sex	Women	216	98	199	93
	Men	4	2	15	7
Age	51–60 years old	57	26	42	20
	41–50 years old	62	28	77	36
	31–40 years old	24	11	64	30
	20–30 years old	77	35	31	15
Education	Vocational secondary	101	46	77	36
	Higher (bachelor’s degree)	95	43	92	43
	Higher (master’s degree)	24	11	45	21
Marital status	Widow/widower	31	14	7	3
	Married	125	57	118	55
	Single	22	10	42	20
	Cohabitation	42	19	47	22
Place of residence	City	117	53	133	62
	Village	103	47	81	38

**Table 2 ijerph-19-05539-t002:** Characteristics of the respondents’ work.

Variable		Nurses		Paramedics	
		N	%	N	%
Work system	Single-shift system (8 h)	37	17	0	0
	12 h shift	183	83	214	100
Professional experience	0–5 years	59	27	30	14
	6–10 years	22	10	25	12
	11–20 years	44	20	57	26
	More than 20 years	95	43	102	48
Affection to the profession	Yes	165	75	110	51
	No	9	4	2	1
	It’s hard to say	47	21	102	48
Workplace	Hospital unit	180	82	15	7
	MCE	31	14	4	2
	Rescue service	7	3	99	46
	ED	2	1	96	45
Number of jobs	1	24	11	167	78
	2	184	84	45	21
	3	12	5	2	1

**Table 3 ijerph-19-05539-t003:** Number of responses given by surveyed nurses and paramedics in each item of Ch. Maslach’s burnout questionnaire.

Items	Nurses		Paramedics	
	YES	NO	YES	NO
Job exhaustion				
I feel emotionally drained because of my work	142	78	104	110
At the end of the workday, I feel worn out	122	98	172	62
Getting up in the morning, I already feel tired when I see a new day’s work ahead of me	142	78	80	134
I feel very exhausted working with people all day	152	68	82	132
My job makes me feel burnt out	58	162	134	80
My job makes me feel frustrated	118	102	102	112
I feel like I’m working too hard	172	48	184	30
I find it too stressful to work in direct contact with people	138	82	66	148
I feel like I’ve reached the limit of my wisdom	72	148	80	134
	1116	864	1004	942
Depersonalization				
I feel like I treat some customers as objects	78	142	64	150
I have become indifferent to people since I have been doing this job	54	166	68	146
I’m afraid my job makes me less compassionate	122	98	84	130
In fact, I don’t care what happens to some clients	56	164	82	132
I feel that some clients and their relatives think that I am responsible for their problems	156	64	106	108
	466	634	404	666
No sense of professional achievements				
I can easily understand what my clients think about certain topics	126	94	122	92
I manage to successfully solve my clients’ problems	78	142	128	86
I feel like I am making a positive impact on lives because of my work	134	86	128	86
I feel a lot of energy inside me	58	162	88	126
I find it easy to create a comfortable atmosphere	84	136	118	96
I feel energized when I work closely with my clients	88	132	116	98
I have achieved many significant goals in my work	104	116	102	112
In my work, I treat emotional problems very calmly	192	124	96	118
	864	992	898	814

**Table 4 ijerph-19-05539-t004:** Overall scores obtained from the Ch. Maslach burnout questionnaire divided into categories.

**Categories of Burnout**	**Nurses/**		**In Total**	**Paramedics**		**In Total**
	**YES**	**NO**		**YES**	**NO**	
EW	1116	864		1004	942	
DP	466	634		404	666	
ZO	864	992		898	814	
	tEW + tDP + nZO/220		11.7	tEW + tDP + nZO/214		10.4
	nEW + nDP + tZO/220		10.7	nEW + nDP + tZO/214		11.7

EW—occupational exhaustion, DP—depersonalization; ZO—No sense of professional achievements.

**Table 5 ijerph-19-05539-t005:** Number of nurses and paramedics surveyed by salary relevance to respondents.

Materiality of Remuneration	Nurses	Paramedics
Yes—the most important thing is job satisfaction	6	4
Rather yes—salary is less critical than overall job satisfaction	18	14
I don’t have a specific opinion about this	12	4
Rather not—I don’t care about my salary	26	34
No—I believe that work should be adequately rewarded	158	158
I am convinced that my salary is not adequate for the work I do (too low)	202	194
I believe my salary is appropriate for the type of work I do	16	20
I think I am rewarded for working good enough	2	0
In total	220	214

**Table 6 ijerph-19-05539-t006:** The correlation between self-reported burnout of paramedics and nurses.

	Self-Assessment of Occupational Burnout		
	YES	NO	*p*
Nurses	12	82	0.000002
Paramedics	82	64	

Significance level *p* < 0.05; results of Pearson’s chi square analysis.

**Table 7 ijerph-19-05539-t007:** The relationship between emotional exhaustion, depersonalization, and occupation of the study groups.

Overall Results of the Ch. Maslach Burnout Questionnaire	Nurses	Paramedics	*p*
Depersonalization	5.1	4.7	0.06
Emotional exhaustion	2.1	1.9	

Significance level *p* < 0.05; results of Pearson’s chi square analysis.

**Table 8 ijerph-19-05539-t008:** The correlation between the occurrence of occupational burnout and the occupation of the study groups.

Overall Results of the Ch. Maslach Burnout Questionnaire	Nurses	Paramedics	*p*
Occupational burnout	11.7	10.4	0.03
Lack of occupational burnout	10.7	11.7	

Significance level *p* < 0.05; results of Student’s *t*-test analysis.

**Table 9 ijerph-19-05539-t009:** The relationship between the length of work experience and physical exhaustion in nurses and paramedics.

Study Group/Variables			Work Experience				
			0–5 Years	6–10 Years	11–20 Years	Over 20 Years	*p*
Nurses	The occurrence of physical exhaustion	Never	0	0	0	0	0.01
		Rarely	26	8	6	18	
		Frequently	28	12	16	62	
		Very frequently	6	2	22	14	
Paramedics	The occurrence of physical exhaustion	Never	2	0	0	0	0.08
		Rarely	8	2	2	18	
		Frequently	14	20	30	56	
		Very frequently	6	4	24	28	

Significance level *p* < 0.05, results of Pearson’s chi square analysis.

**Table 10 ijerph-19-05539-t010:** Correlation between salary relevance for nurses and paramedics.

		Nurses	Paramedics	*p*
Relevance of remuneration	Yes—the most important thing is job satisfaction	6	4	0.57
	Rather yes—salary is less important than overall job satisfaction	18	14	
	I don’t have a specific opinion about this	12	4	
	Rather not—I don’t care about my salary	26	34	
	No—I believe that work should be adequately rewarded	158	158	

Significance level *p* < 0.05; results of Pearson’s chi square analysis.

**Table 11 ijerph-19-05539-t011:** The correlation between salary adequacy of nurses and paramedics.

		Nurses	Paramedics	*p*
**Adequacy of remuneration**	I am convinced that my salary is not adequate for the work I perform (it is too low)	202	194	0.53
	I believe my salary is adequate for the type of work I perform	16	20	
	I think I am rewarded for my work good enough	2	0	

Significance level *p* < 0.05; results of Pearson’s chi square analysis.

**Table 12 ijerph-19-05539-t012:** The correlation between the use of psychological help and dealing with death.

		Use of Psychological Support		
		YES	NO	*p*
**Stress factor: exposure to death in a given study group**	Nurses	4	44	0.93
	Paramedics	12	122	

Significance level *p* < 0.05; results of chi square analysis.

## Data Availability

The data presented in this study are available on reasonable request from the corresponding author.

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
