# Peer review of "Stress-Inducing Factors vs. the Risk of Occupational Burnout in the Work of Nurses and Paramedics"

_ijerph, 2022, doi:10.3390/ijerph19095539_

Round 1
Reviewer 1 Report
I am wondering about the reasoning behind merging paramedics and nursing side by side, would it have been easier to connect emergency nursing and paramedics to at least create a connection between the two or was using nursing just a broad term? It may be helpful to consider this from one lens such as emergency medicine.
Line 67/68 - you identify a nurse a she, might consider a more gender neutral term here if you're not speaking about a specific nurse
Line 114 - 2.1 - the demographic of the nurses and paramedics may throw off the numbers because you could have obtained data from nurses that don't work in emergency medicine and their stress level might be different and may look more like burnout due to long term suffering as opposed to the high stress trauma induced environment of emergency medicine.
Line 131 - utilization of the word proves seems quite bold, might consider something like suggests that gender might be a factor in analysis
Line 148 - mention seeking services from a psychologist - are psychologists the only mental health professionals available? Are there any seeing psychiatry or family medicine and on antidepressants or anti-anxiolytics to manage stress and anxiety? Could you also have people seeing masters level clinicians like social workers, marriage and family therapists and so on?
Line 153 - by students do you mean participants? Do you call participants students in the Helsinki Declaration?
Line 197 - you may not see a correlation because of a confounding variable, you might not be correlating like with like such as paramedics and emergency room nurses. You might be getting mixed and confused data because paramedics might be getting correlated with oncology, cardio, family medicine, internal medicine, or surgical nursing which would illicit different responses to burnout.
Line 265 - the word prove again looks strong might consider suggests or correlates as this is not a double blind study that can show proof.
Discussion section - seems a bit mixed up you have a discussion comparing and contrasting your teams results to that of other results. It may be more helpful to separate these sections and create a future research or correlated research section and have the comparison or contrast be done in that section
Line 396 - Limitation of the study - another limitation is the breath of the nursing field that could create a compounding factor. In the demographic section you do not highlight what areas of medicine these nurses are coming from and the clinic setting versus the emergency department setting might have different levels of burn out so you might not be comparing in similarity.
Author Response
Dear Reviewer,
Thank you very much for your valuable comments, we hope that all the changes made will be satisfactory.Authors

Reviewer 2 Report
this manuscript is a written piece of paper in a well-organized form. I didn't observe any major issue or deficiency that needed to be addressed. The introduction is fluent and offers an adequate discussion of the problem and its background. The methodology section is also well-organized and provides sufficient information on the population characteristics, size, sampling and data analyses techniques. One minor issue- under the heading 2.4 "ethical consideration" the term "student" is confusing. the author may provide an explanation of this term has been used as this study was undertaken by nurses and paramedics. further, Figures 1 and 2, with descriptions should be better put in the result section. in the result section, the author has provided the p values with correlation scores, but with no discussion on the test applied for it. better to provide discussion in a few lines on the test adopted for calculating the p values for clarity and verification. lastly, the discussion section is fine a d contains some citations to the latest studies. however, could be better to provide a few s]relevent studies if existed from 2020, 2021, 2022 etc.
Author Response
Dear Reviewer,
Thank you very much for your valuable comments, we hope that all the corrections we have introduced will be satisfactory.Authors

Round 2
Reviewer 1 Report
Section 2.4 Ethical Considerations (Line144) - is this specific to nurses or do you mean the subjects of the study were all informed of the confidentiality...? May want to consider a word change to include all the subjects.
Author Response
Dear Reviewer,
Thank you very much for your valuable comments and we apologize for an oversight. All corrections have been marked in the text.Autors